# Study on the Ground-Penetrating Radar Response Characteristics of Pavement Voids Based on a Three-Phase Concrete Model

**DOI:** 10.3390/s25185713

**Published:** 2025-09-12

**Authors:** Shuaishuai Wei, Huan Zhang, Jiancun Fu, Wenyang Han

**Affiliations:** 1School of Civil Engineering, Qingdao University of Technology, Qingdao 266520, China; shuaishuai_wei@163.com; 2Shandong Transportation Institute, Jinan 250102, China; fujiancunhit@163.com (J.F.); hanwenyang@sdjtky.cn (W.H.)

**Keywords:** three-phase concrete pavement model, void detection, ground penetrating radar, aggregate scattering, FDTD simulation

## Abstract

Concrete pavements frequently develop subsurface voids between surface and base layers during long-term service due to cyclic loading, environmental effects, and subgrade instability, which compromise structural integrity and traffic safety. Ground-penetrating radar (GPR) has been widely used as a non-destructive method for detecting such voids. However, the presence of coarse aggregates with strong electromagnetic scattering properties often introduces pseudo-reflection signals in radar images, hindering accurate void identification. To address this challenge, this study develops a high-fidelity three-phase concrete model incorporating aggregates, mortar, and the interfacial transition zone (ITZ). The Finite-Difference Time-Domain (FDTD) method is used to simulate electromagnetic wave propagation in both voided and intact structures. Simulation results reveal that aggregate-induced scattering can blur or distort reflection interfaces, generating pseudo-hyperbolic anomalies even in the absence of voids. In cases of thin-layer voids, real echo signals may be masked by aggregate scattering, leading to missed detections. GPR systems can be broadly classified into impulse, continuous-wave, and multi-frequency types. To validate the simulations, field tests using multi-frequency 2D/3D GPR systems and borehole verification were conducted. The results confirm the consistency between simulated and actual radar anomalies and validate the proposed model. This work provides theoretical insight and modeling strategies to enhance the interpretation accuracy of GPR data for subsurface void detection in concrete pavements.

## 1. Introduction

With the rapid advancement of urbanization, road networks in medium and large cities have become increasingly dense, and concrete pavements are widely utilized in municipal roads and critical transportation hubs [1,2,3]. During service, urban concrete pavements are prone to developing subsurface defects, such as voids and cracks, at the interface between the surface and base layers due to repeated traffic loads and environmental factors, including temperature and moisture variations [4,5,6,7]. These voids result from a combination of factors, including insufficient base compaction, underground utility leakage, soil erosion, freeze–thaw cycles, fatigue damage from traffic loads, and differential settlement of the subgrade. Once formed, voids undermine the support relationship between the pavement slab and the base layer, leaving the structure in a state of suspended loading. This not only accelerates water ingress and pumping effects but also intensifies bottom erosion and cracking, ultimately leading to local settlement or even large-scale collapse [8,9,10]. If such voids are not detected and addressed early, they may rapidly evolve into irreversible structural failures with potentially catastrophic impacts on urban operations. Hence, developing an efficient, accurate, and non-destructive detection and assessment system has become an urgent need for urban pavement maintenance [11].

To date, a variety of physical and geophysical methods have been established for subsurface void detection, including acoustic methods [12,13,14], Distributed Optical Fiber Vibration Sensing (DOVS) [15], Falling Weight Deflectometers (FWD) [16], Dynamic Cone Penetration tests (DCP) [17], and Ground Penetrating Radar (GPR) [3,18,19,20]. Although acoustic methods, DOVS, FWD, and DCP can reveal subsurface conditions to some extent, they generally suffer from low efficiency, complex operation, and limited adaptability. GPR, on the other hand, has emerged as a powerful tool for void detection due to its non-contact nature, high resolution, and real-time imaging capabilities [21]. Depending on the type of emitted signal, GPR systems can be generally divided into three classes: (i) impulse radars, which emit short pulses typically similar to a single period of a sinusoid; (ii) continuous-wave radars with varying frequency laws; and (iii) holographic radars, which are usually multi-frequency and designed to produce three-dimensional subsurface images [22,23,24,25,26]. Impulse GPR has been widely employed in road engineering due to its relatively simple system structure and strong adaptability, whereas holographic radars have demonstrated superior imaging resolution and recognition capabilities, enabling fine-scale subsurface imaging even under complex conditions [23,24,25,26]. Recent advances in synthetic spectrum imaging and back-projection algorithms further enhance holographic GPR performance, showing high resolution and fast computation [25]. Research on the application of GPR in non-destructive testing (NDT) of roads and concrete structures has yielded substantial progress, covering structural identification, damage detection, parameter inversion, and intelligent data processing. For concrete structures, GPR is widely used in rebar detection, void identification, and structural evaluation due to its high resolution and non-destructive nature. For instance, studies have summarized typical GPR practices for reinforced concrete detection, showing that dual-polarization antennas combined with migration processing significantly improve the accuracy of rebar recognition [27]. Other works have investigated how antenna frequency and void geometry affect GPR signal characteristics, offering guidance for selecting detection parameters [28]. Moreover, the integration of GPR and eddy current testing has revealed the influence of reinforcement layout and concrete cover thickness on crack propagation, demonstrating the technique’s potential in analyzing the durability of continuously reinforced concrete pavements [29].

In the context of asphalt concrete pavements, GPR is primarily used to detect density, moisture content, cracks, and the condition of structural layers. Some studies have developed numerical models based on three-phase dielectric theory and validated them through field testing to analyze the relationship between pavement density and dielectric properties [30]. Filtering algorithms have also been proposed to mitigate noise caused by antenna vibrations, thereby enhancing real-time monitoring capabilities during construction [31]. For moisture monitoring, numerical models have been developed under varying saturation levels, with empirical formulas proposed to relate dielectric constant and volumetric water content [32]. In terms of structural integrity assessment, preliminary systems have been established using three-dimensional (3D) GPR for disease feature recognition and the Pavement Internal Condition Index (PICI) [33]. However, the heterogeneous nature of pavement materials and environmental interference limits the applicability of numerical models, and existing filtering techniques often struggle to address strong noise interference.

The development of 3D GPR in recent years has provided a technical breakthrough for high-precision identification of subsurface defects in pavement structures. Unlike traditional two-dimensional (2D) radar, which only captures vertical profile images, 3D radar employs antenna arrays to collect large-scale, high-resolution volumetric data in real-time during vehicle motion, enabling the 3D visualization of structural integrity and damage features. Depending on the coupling method between the antenna and pavement surface, 3D GPR systems are classified into air-coupled and ground-coupled types. Air-coupled antennas are easy to deploy and suitable for rapid, large-area surveys—such as evaluating pavement layer thickness and structural uniformity—but their limited penetration depth and spatial resolution, due to air gaps, hinder the identification of deeper or smaller-scale anomalies. In contrast, ground-coupled antennas enhance signal-to-noise ratio (SNR) by maintaining contact with the surface and supporting higher-frequency operation. They are more suitable for detecting surface cracks, shallow voids, and delamination, and are widely used for high-precision damage localization and assessment [34,35,36]. Recently, air-coupled horn antennas have been widely adopted for non-destructive electromagnetic testing of pavement structures, primarily for assessing interlayer interfaces [37,38,39], compaction quality [40,41], moisture content in soil and subgrade [42], and for analyzing dielectric and mechanical properties of materials [43]. However, due to their limited sampling rate, air-coupled antennas exhibit constrained spatial resolution under high-speed operation, making it difficult to identify small-scale or poorly defined defects such as base-layer voids or localized delamination. In contrast, ground-coupled antennas offer higher spatial resolution and SNR, making them better suited for detecting shallow subsurface discontinuities [44]. High-frequency ground-coupled antennas can effectively detect small-scale defects in concrete structures, including cracks, repairs, and localized voids; however, their performance is significantly influenced by material properties, target size, dielectric contrasts, and interface morphology [45]. Therefore, accurately simulating GPR electromagnetic responses to pavement voids requires realistic structural modeling and high-resolution antenna parameters.

In summary, current GPR techniques still face numerous challenges in detecting subsurface defects in concrete pavements, particularly in identifying base-layer voids, filtering pseudo-reflections, and resolving fine, shallow features. Conventional homogeneous models fail to accurately represent the true multiphase structure of concrete, resulting in substantial deviations in wave propagation paths, reflection patterns, and signal interpretations, thereby compromising diagnostic accuracy. To address these issues, this study proposes a multiphase concrete model based on the composition of aggregate, mortar, and interfacial transition zone (ITZ). Various void types and structural parameters are modeled to simulate electromagnetic wave propagation using the Finite-Difference Time-Domain (FDTD) method [46,47]. In this study, the FDTD method was chosen as the core numerical simulation approach for the following key reasons: First, most commercial 2D and 3D GPR systems, such as the 3D-Radar and ImpulseRadar used in this study, perform data acquisition and processing in the time domain. Therefore, employing the FDTD method ensures better alignment with practical engineering applications. Second, the FDTD method directly solves Maxwell’s equations and maintains high accuracy and stability, even in complex heterogeneous media, allowing for effective characterization of electromagnetic wave propagation, reflection, and scattering in various geological materials. Additionally, the FDTD method offers the advantage of explicit time-domain computation. It provides complete time-series data. This data not only supports detailed analysis of GPR signal reflections but also lays a solid foundation for subsequent radar waveform feature analysis. Thus, the adoption of the FDTD algorithm is necessary both to maintain consistency with existing radar systems and to study the electromagnetic response behavior of heterogeneous media. The study systematically investigates how aggregate heterogeneity distorts GPR signals and contributes to the formation of pseudo-anomalies. By comparing B-scan responses from 11 representative models, the research identifies distinct scattering patterns associated with aggregates versus true void features. Furthermore, field data collected using multi-frequency GPR systems are used to validate simulation results. The findings offer theoretical insights and modeling strategies for improving the accuracy of void detection in concrete pavements, thereby opening new avenues for GPR data interpretation and engineering defect diagnosis.

## 2. FDTD Numerical Simulation

In 2D FDTD simulations [46,47], the most commonly employed configuration is the transverse magnetic (TM) mode, where the electric field is oriented along the z-direction, and the magnetic fields are distributed along the x- and y-directions. This mode is well-suited for modeling wave propagation problems in 2D cross-sectional domains. Assuming that the electromagnetic fields vary only in the *x*–*y* plane, with the electric field containing only the *E_z_* component and the magnetic field comprising *H_x_* and *H_y_*, Maxwell’s equations can be simplified as follows:(1)$$\begin{array}{l} \frac{\partial H_{y}}{\partial x}-\frac{\partial H_{x}}{\partial y}=\varepsilon \frac{\partial E_{z}}{\partial t}\\ \frac{\partial E_{z}}{\partial y}=-\mu \frac{\partial H_{x}}{\partial t}\\ -\frac{\partial E_{z}}{\partial x}=-\mu \frac{\partial H_{y}}{\partial t} \end{array}$$
where *E_z_* is the out-of-plane electric field component (V/m), *H_x_* and *H_y_* are the magnetic field components along the *x* and *y* directions (A/m), ε is the permittivity (F/m), and *μ* is the permeability (H/m).

The key procedure in the FDTD method is the discretization of the partial differential equations in both the spatial and temporal domains. This involves approximating continuous variables on a discrete grid, enabling numerical time-stepping iteration. To numerically solve Maxwell’s equations on a computer, the computational domain must be discretized spatially and temporally.

For 2D GPR problems, the computational domain is typically defined as a rectangular area in the *x–y* plane, with dimensions *L_x_ × L_y_*. This region is uniformly discretized into a grid with *N_x_* and *N_y_* nodes along the *x* and *y* directions, respectively. The spatial steps are $\Delta x=L_{x}/N_{x}$ and $\Delta y=L_{y}/N_{y}$, and each grid point is identified by the index (*i*, *j*), corresponding to the spatial location (i⋅Δx, j⋅Δy). The total simulation time T is divided into *N_t_* steps of size $\Delta t=T/N_{t}$, and the electromagnetic field values are iteratively updated at each time step *n*.

To ensure symmetry and numerical stability, the electric and magnetic field components are spatially staggered on a Yee grid [48]. Specifically, the electric field component *E_z_*, which is normal to the *x–y* plane, is located at the primary grid point (*i*, *j*). The magnetic field component *H_x_* is located at (*i*, *j* + 1/2), i.e., between the top and bottom of the *E_z_* point, while *H_y_* is located at (*i* + 1/2, *j*), i.e., between the left and right of the *E_z_* point. Applying central difference approximations to the spatial and temporal derivatives, the update equations for the TM mode in 2D are derived as:(2)$$\begin{array}{l} E_{z}^{n+1}(i,j)=E_{z}^{n}(i,j)+\frac{\Delta t}{\varepsilon (i,j)}\left[\frac{H_{y}^{n+1/2}(i,j)-H_{y}^{n+1/2}(i-1,j)}{\Delta x}-\frac{H_{x}^{n+1/2}(i,j)-H_{x}^{n+1/2}(i,j-1)}{\Delta y}\right]\\ H_{x}^{n+1/2}(i,j)=H_{x}^{n-1/2}(i,j)-\frac{\Delta t}{\mu (i,j)}\cdot \frac{E_{z}^{n}(i,j+1)-E_{z}^{n}(i,j)}{\Delta x}\\ H_{y}^{n+1/2}(i,j)=H_{y}^{n-1/2}(i,j)+\frac{\Delta t}{\mu (i,j)}\cdot \frac{E_{z}^{n}(i+1,j)-E_{z}^{n}(i,j)}{\Delta x} \end{array}$$

These equations are iteratively solved over time to simulate the dynamic propagation of electromagnetic waves in the 2D domain. To ensure numerical stability, the FDTD method must satisfy the Courant–Friedrichs–Lewy (CFL) condition [49], which constrains the maximum allowable time step:(3)$$\Delta t\leq \frac{1}{c\sqrt{\frac{1}{\Delta x^{2}}+\frac{1}{\Delta y^{2}}}}$$
where *c* is the speed of electromagnetic waves in the medium. This condition is closely tied to spatial discretization and the properties of the medium.

In practice, as the computational domain is finite, absorbing boundary conditions must be applied to prevent artificial reflections from the edges. The most widely used method is the perfectly matched layer (PML) [50], which effectively absorbs outgoing waves without reflecting them into the computational region. Unlike earlier absorbing boundary conditions [51], the PML can theoretically achieve reflection-free absorption for electromagnetic waves incident at arbitrary angles and frequencies. This property is crucial for GPR simulations where scattered waves from aggregates and void interfaces impinge on the domain boundaries over a wide range of directions. By using PML, spurious boundary reflections are minimized, ensuring that the recorded signals originate only from the modeled pavement structure. The specific FDTD process is described in Figure 1.

## 3. Concrete Pavement Structure and Modeling

Concrete pavement is a type of rigid pavement widely used in urban roads, highways, airport runways, and industrial infrastructure. Compared with asphalt pavement, it offers several advantages, including higher strength, better durability, lower deformation, and longer maintenance intervals. Concrete pavements are typically composed of multiple layers. Each layer has a distinct function and material composition, working together to carry traffic loads and resist environmental effects. A typical structure consists of three components: a surface layer, a base layer, and a subgrade. The surface layer is typically composed of cement concrete slabs and serves as the uppermost part of the pavement. It directly bears the loads from vehicles and environmental influences. The material is ordinary or high-performance concrete, mainly composed of aggregates, cement, and water. The thickness generally ranges from 18 cm to 30 cm. This layer has high strength and stiffness, serving as the primary load-bearing layer. The base layer lies beneath the surface layer. It helps distribute loads and improves the overall structural stability. Common materials include cement-stabilized crushed stone, graded gravel, or dry lean concrete. The aggregates in this layer are coarser, and the structure is relatively loose but of moderate strength. The thickness typically ranges from 15 cm to 25 cm. The bottom layer is the subgrade, which consists of natural soil and supports the entire pavement structure.

This study focuses on the debonding between the surface and base layers. To investigate this, a 2D model was developed as shown in Figure 2. Both the surface and base layers were set to a thickness of 20 cm. In this chapter, the concrete pavement is modeled as a three-phase composite consisting of aggregates, ITZ, and mortar. Based on actual 2D aggregate images, the aggregate structures in the surface and base layers are constructed using a polygonal approximation method [47]. This method offers several advantages. First, the aggregate shapes can be flexibly controlled by adjusting polygonal vertices, which provides a closer representation of the irregular geometries of real aggregates compared to circular or elliptical simplifications. Second, the approach is highly parameterized, allowing aggregate size, orientation, and distribution to be statistically defined, which enhances the randomness and representativeness of the model. Third, when combined with actual 2D aggregate images, it achieves high modeling accuracy by reproducing the spatial heterogeneity of concrete microstructure. These features make the polygonal approximation particularly suitable for defining the spatial distribution of dielectric constants in FDTD-based GPR simulations [47].

### 3.1. Simulation of Concrete Aggregates

To better represent the internal structure of real concrete, this study identifies the 2D contours of aggregates based on images of actual concrete slices. Figure 3 presents a representative image showing the outlines of aggregates, marked in red. As seen in the figure, the distribution of aggregates is random, and their geometries are highly irregular. Most aggregates can be approximated as irregular closed polygons, commonly with 4 to 6 sides. Such geometric characteristics are typical of natural gravel and crushed stone aggregates. These shapes are neither circular nor elliptical, and often feature sharp angles and jagged boundaries.

To accurately simulate the spatial distribution and geometric features of coarse aggregates in concrete, a large number of irregularly shaped aggregates must be generated. Traditional 2D random aggregate generation methods often rely on distance-based exclusion criteria to avoid overlapping between new and existing aggregates. However, as the number of aggregates increases, these methods suffer from low efficiency and poor convergence. To overcome these limitations, this study introduces the Fast Grid Region Division Method (FGRDM) [47]. This method significantly improves aggregate generation efficiency and is particularly suitable for modeling high-volume-fraction concrete structures. During the modeling process, different size distributions are assigned to aggregates in the surface and base layers. Aggregates in the surface layer are smaller, while those in the base layer are relatively larger. The shapes are mainly modeled as quadrilaterals, pentagons, and hexagons. Randomization parameters include the center position, rotation angle, scaling factor, and boundary perturbation. These parameters simulate realistic particle size distributions and spatial arrangements. The resulting 2D aggregate models preserve key geometric features from actual images, improve spatial accuracy in electromagnetic simulations, and enhance structural realism. This provides a reliable foundation for radar reflection analysis and damage identification.

The core idea of FGRDM is to divide the unoccupied domain into grids. Each grid is assigned a value *d*, representing the maximum allowable aggregate size (equivalent to the radius of the largest inscribed circle). For a candidate aggregate with radius *H*, one grid cell is randomly selected as the aggregate center from all grids where *d* ≥ *H*. Once a new aggregate is generated, the occupied region is excluded from subsequent calculations. This effectively avoids overlapping. As more aggregates are generated, the available area decreases, but the generation speed increases due to the reduced number of candidate regions, thereby improving the overall modeling efficiency. This method supports both regular (e.g., ellipses) and irregular (e.g., polygons with perturbed edges) aggregate generation. It can also be extended to construct ITZ between aggregates and mortar. Compared with traditional methods, this approach ensures non-overlapping placement, accelerates generation, and improves model controllability. It provides an efficient and reliable geometric basis for electromagnetic modeling and FDTD-based ground-penetrating radar simulations of concrete. Figure 4 shows simulated aggregates with different shapes. The simulated results closely resemble the actual aggregate geometries shown in Figure 3.

### 3.2. Concrete Pavement Structure

During the modeling process, the aggregate models generated in the previous section are embedded into the surface and base layers, respectively. The distinction between the surface and base layers is made primarily due to significant differences in material composition, aggregate size, and structural function. The surface layer typically uses finer aggregates with dense gradation to improve surface smoothness and wear resistance, while also bearing vehicle loads and environmental impacts. In contrast, the base layer often uses coarser aggregates to ensure structural strength and load-bearing capacity. These differences in particle size and gradation directly affect the porosity, moisture content distribution, and electromagnetic wave propagation characteristics of the concrete, leading to variations in electromagnetic parameters such as dielectric constant and conductivity between the surface and base layers. Therefore, to more accurately reflect the pavement’s electromagnetic response, a more refined representation of both the surface and base layers is employed in the modeling process [52,53]. Each region is assigned corresponding electromagnetic parameters based on its material characteristics.

To reflect realistic engineering conditions, aggregate placement adheres to the mix design requirements for concrete. Different particle size distributions are used for the surface and base layers. In the surface layer, fine aggregates dominate. According to the mix design, aggregates sized 0–5 mm and 5–10 mm are added at 600 kg/m^3^ and 141 kg/m^3^, respectively. Coarser aggregates of 10–20 mm (515 kg/m^3^) and 20–25 mm (752 kg/m^3^) are also included (Table 1). The admixture is PCA-I type high-efficiency water reducer. This combination enhances surface smoothness and concrete compactness, as shown in Figure 5. In contrast, the base layer utilizes larger aggregates to enhance its mechanical performance and load-bearing capacity. Aggregates sized 20–30 mm and 10–20 mm account for 18% and 40% of the total, respectively. Smaller aggregates, ranging from 5–10 mm (15%) to 0–5 mm (27%), are also included (Table 2). Figure 6 illustrates the resulting base layer structure. Based on the aggregate gradation in each structural layer, corresponding geometric aggregate models are generated and embedded into the surface and base layer regions. This approach enables a more realistic representation of the material composition in each layer.

To ensure simulation accuracy and numerical stability, the model uses a uniform grid, with each cell assigned specific material properties. The dielectric constants for aggregates, cement mortar, and potential ITZ are defined separately (Table 3), resulting in a spatially heterogeneous dielectric distribution. Perfectly matched layer (PML) boundary conditions are applied at the model boundaries to eliminate artificial reflections during radar wave propagation. The core objective of this structural model is to create a computational domain that captures both the realistic geometry and dielectric heterogeneity of concrete. This provides a physically accurate foundation for analyzing the influence of interface conditions between the surface and base layers, such as debonding, on the propagation of electromagnetic waves.

## 4. Electromagnetic Field Simulation of Different Concrete Pavement Models

To investigate the propagation behavior of GPR electromagnetic waves in concrete subgrades with irregular aggregate structures, a series of numerical models was developed using the FDTD method [46,47]. The simulation domain consists of 450 × 800 grid points, with a spatial resolution of 0.001 m. The time window is set to 15 ns. Perfectly matched layer (PML) boundary conditions are applied to eliminate artificial reflections. A 0.05 m-thick air layer is placed above the model. The source is a Ricker wavelet with a central frequency of 1 GHz and a peak amplitude of 1, polarized in the vertical (*z*) direction. The transmitting and receiving antennas move synchronously along the horizontal axis with a 2 mm step, generating a typical B-scan image. In B-scan images, the horizontal axis represents the movement of the antenna along the survey line, and the vertical axis corresponds to the two-way travel time of radar signals, which can be converted to depth. The amplitude of the reflected signal is shown in grayscale or pseudo-color. Subsurface anomalies such as interfaces, voids, or inclusions cause abrupt changes in electromagnetic properties, resulting in strip-like or hyperbolic reflections in the B-scan image. These features provide essential clues for target identification and defect analysis.

A total of 11 representative concrete subgrade models were constructed to simulate GPR responses under various structural configurations and voiding conditions. Model 1 assumes a homogeneous concrete structure in both the surface and base layers (Figure 7a). It serves as the reference case for analyzing wave propagation, attenuation, and reflection under ideal, uniform conditions. Model 2 introduces a void in the base layer, maintaining material homogeneity (Figure 8a). This configuration helps isolate the effects of voiding on reflection strength and time delay. Model 3 incorporates realistic concrete heterogeneity by including coarse aggregates, mortar, and an ITZ, forming a three-phase medium (Figure 9a). This model simulates scattering, velocity changes, and energy loss induced by internal heterogeneities. It also aids in distinguishing false reflections caused by material non-uniformity, improving the accuracy of void detection. Models 4 and 5 introduce voids with thicknesses of 5 mm and 10 mm, respectively, to examine how thin-layer voids affect reflection amplitude and detectability (Figure 10). Models 6 and 7 are designed to assess the role of void continuity (Figure 11a and Figure 12a). Model 6 simulates a continuous strip-shaped void, highlighting energy superposition and interference from aligned reflections. Model 7 represents irregular or isolated voids, capturing scattering and diffraction effects that are typical of localized defects. Models 8, 9, and 10 simulate voids with thicknesses of 5 mm, 10 mm, and 15 mm, respectively (Figure 13a, Figure 14a and Figure 15). These configurations aim to evaluate the detectability and resolution limits of a 1 GHz radar system under heterogeneous conditions, especially for ultra-thin voids. Finally, Model 11 introduces water as a high-permittivity filler within the void to simulate moisture accumulation (Figure 16a). This scenario reflects strong signal responses and multiple echoes, offering insight into water-induced anomalies such as leakage or structural deterioration.

By comparing Figure 7, Figure 8 and Figure 9, it is evident that the presence of aggregates can induce pseudo-anomalous hyperbolic reflections. In Figure 7, prominent hyperbolic waveforms appear at both ends of the void region. Furthermore, small-scale interlayer defects, such as moisture accumulation or microcracks, may also enhance radar echo intensity. However, such responses do not necessarily indicate the presence of actual voids. It is important to note that enhanced radar amplitude at interlayer interfaces does not always signify voiding. The observed increase may result from constructive interference or reflection amplification at material boundaries with differing electromagnetic properties. Therefore, void detection should be based on a comprehensive analysis that includes the shape of hyperbolic reflections, their spatial continuity, and frequency-dependent characteristics. Figure 17 presents the B-scan results corresponding to the two models in Figure 10. The left and right sections represent void thicknesses of 5 mm and 10 mm, respectively, in three-phase concrete models. Although the voids have the same horizontal extent of 100 mm, the 5 mm void produces a weaker reflection due to aggregate-induced scattering, and its hyperbolic reflection feature is not clearly visible. In contrast, the 10 mm void shows a significantly stronger echo. This suggests that aggregates can reduce radar sensitivity to thin-layer voids, potentially leading to the misidentification or omission of shallow defects during field detection. In the continuous void model shown in Figure 11, two distinct hyperbolic reflections appear in the B-scan image. These reflections are primarily associated with the boundary effects on the left and right sides of the upper void surface, which is consistent with the lateral effect of the void surface. In the discontinuous void model illustrated in Figure 12, each void has two lateral boundaries that may give rise to hyperbolic reflections, theoretically resulting in four hyperbolas. However, the simulation results reveal only two prominent hyperbolic reflections. This discrepancy can be attributed to the limitation of lateral resolution, which prevents the radar from fully separating reflections from adjacent boundaries. Therefore, we infer that the two hyperbolas in Figure 11 indeed correspond to the lateral effects of the void boundaries, while the results in Figure 12 further highlight the significant influence of limited lateral resolution on the visibility of hyperbolic reflections. This finding also underscores the necessity of cautious interpretation of boundary features in small-scale or closely spaced defects during practical engineering inspections. The analysis of Figure 12 and Figure 13 reveals that a thin void layer located between two adjacent voids can cause superimposed reflections. This effect results in strong horizontal echoes in the B-scan, which obscure the original hyperbolic features and reduce their visibility. The comparison between Figure 13 and Figure 14 shows that the 10 mm void produces stronger reflections than the 5 mm void. Due to aggregate scattering and signal attenuation, thin-layer voids can be easily misinterpreted as natural interlayer reflections in practice, making them difficult to identify accurately. Figure 15 shows the B-scan of a three-phase concrete model with a 15 mm void. The reflected signal from the void is significantly stronger. Figure 18 displays the B-scan results for models nine and ten. Both the 10 mm and 15 mm voids exhibit clearly identifiable thin-layer reflections. The reflection associated with the 10 mm void also presents local phase discontinuity, suggesting interface perturbations. Figure 16 corresponds to model eleven, which includes a water-filled void. Compared to model ten, the electromagnetic impedance contrast is more pronounced due to the high dielectric constant of water. As a result, the reflection amplitude is significantly enhanced, and multiple reflections are clearly observed. These results confirm that water-filled voids exhibit stronger electromagnetic responses, supporting their detectability in practical applications.

## 5. Validation with Field Data

To assess the effectiveness and practical applicability of the simulation results, field tests were conducted. Two representative GPR systems were selected for joint evaluation. The first system is the GeoScope™ MK IV 3D GPR (3D-Radar A/S, Trondheim, Norway), developed by 3D-Radar, equipped with a DXG1820 ground-coupled antenna array (Figure 19a). The second is the CO1760 dual-frequency 2D GPR system from ImpulseRadar (Figure 19b). The two systems offer complementary technical advantages. GeoScope™ MK IV enables rapid, large-area scanning and structural imaging. CO1760, in contrast, provides high-resolution detection. Their combined use facilitates multi-scale and multi-depth characterization of subsurface features and anomalies.

GeoScope™ MK IV employs stepped-frequency continuous wave (SFCW) technology. It transmits a wide range of frequencies (200 MHz to 3 GHz) via a digital signal source. The DXG1820 antenna array includes 21 pairs of electronically scanned bow-tie antennas. With a channel spacing of 75 mm and a total scan width of 1.5 m, the system achieves high-density data acquisition in a single scan (Table 4). It offers both shallow, high-resolution imaging and deep-penetration capabilities. This system has demonstrated excellent performance in detecting road layers and bridge deck defects, providing continuous and clear 3D radar images that facilitate the rapid identification of subsurface variations.

The CO1760 system integrates two ground-coupled antennas with central frequencies of 170 MHz and 600 MHz. It operates using continuous waveform (CWF) technology, delivering high time resolution and signal-to-noise ratio. The low-frequency channel ensures sufficient penetration depth for deep structures, while the high-frequency channel offers finer resolution for detecting shallow features such as voids and layer separations. It is particularly suitable for non-destructive testing of road structures, underground utilities, and municipal infrastructure.

In this study, GeoScope™ MK IV was first used for the rapid scanning of the test area to establish an overall subsurface profile and to locate suspicious anomalies preliminarily. Detailed follow-up surveys were then performed using the CO1760 system to enhance resolution and analyze anomaly characteristics. The combined interpretation of dual-frequency data significantly improved anomaly recognition. It was particularly effective for identifying delaminations, voids, and pseudo-reflections.

Special attention was paid to strong scattering signals that appeared similar to voids in the radargrams. Further analysis revealed that these reflections were often attributed to coarse aggregate distributions, discontinuities in the interfacial transition zone, or dielectric heterogeneity within the concrete. These are classified as pseudo-anomalies rather than true structural defects. Without cross-verification using multi-frequency responses, they could easily be misinterpreted. Therefore, systematic interpretation of waveform characteristics, spatial patterns, and frequency-dependent responses is essential. This approach helps to eliminate misleading signals and improve the accuracy and reliability of subsurface defect diagnosis.

To visualize the lateral distribution of subsurface features at different depths, C-scan images were generated using multi-frequency GPR data. Compared to traditional B-scans (vertical sections), C-scans are more suitable for capturing the horizontal extent and pattern of shallow anomalies. This enables improved detection of defects, such as voids and delaminations, thereby enhancing lateral diagnostic accuracy in concrete structures.

Figure 20 shows on-site photographs of the 3D and 2D radar systems during field surveys. The GeoScope™ MK IV system, mounted on a vehicle, was employed for rapid large-area scanning, especially suited for urban roads. Due to heavy traffic, data collection was scheduled at night. The survey vehicle operated at a controlled speed of approximately 20 km/h to minimize interference with regular traffic.

During the initial survey, an area with abnormal reflections was identified (Figure 21). The C-scan from the 3D GPR (Figure 21b) revealed continuous and strong reflections across all antenna channels. This anomaly exhibited significant lateral continuity and high reflection intensity, suggesting the potential presence of structural damage or a void.

To further examine this area, the ImpulseRadar CO1760 system was used for high-resolution resurvey. Figure 22 shows the B-scan image from the 2D radar, which captured strong reflections with waveform characteristics consistent with those observed in Figure 21a, confirming the presence of a structural anomaly. The 2D GPR image used in this study was obtained using a 600 MHz antenna, which is suitable for high-resolution imaging of shallow subsurface structures. The data were processed using Reflexw software (version 7.0), a widely used tool for GPR data analysis. In the data processing stage, a 1D filter was applied to enhance the signal quality. Specifically, the subtract-DC-shift filter was used to remove low-frequency noise and to shift the DC component, ensuring that the data reflected more accurate and meaningful subsurface features. This filtering technique is particularly effective for removing baseline drift and enhancing the clarity of reflections from subsurface structures. In both Figure 21a and Figure 22, hyperbolic reflections at the void boundaries are clearly observed, showing strong consistency with the simulation results. This agreement not only demonstrates the capability of the simulations to reproduce real-world conditions but also highlights the effectiveness of ground-penetrating radar in identifying void boundaries, providing solid support for defect detection in engineering practice. A follow-up drilling investigation was performed to validate the radar results. As shown in Figure 23, a void was confirmed at the identified location. The void was located approximately 0.22–0.31 m below the road surface, with an estimated lateral extent of 2 m × 22 m. It was found near a maintenance cover and close to a joint between concrete slabs. Minor surface cracking was observed in the area. The anomaly likely resulted from water infiltration through surface cracks, which caused the progressive erosion of the backfill under repeated heavy traffic loads, eventually forming a cavity above the pipeline.

During the propagation of high-frequency electromagnetic waves, the triphasic composite structure of concrete induces strong scattering, diffraction, and interference. These effects result in significant random noise and distortions in the radar profiles, which complicate the identification of actual structural interfaces. Numerous hyperbolic reflections, frequently observed in the radargrams, are typically pseudo-anomalies. These are caused not by structural defects but by localized scattering from randomly distributed coarse aggregates of varying size.

In some regions, reflections from the interface between surface and base layers appear indistinct. This is mainly due to two reasons: (1) similar dielectric constants between the two layers result in weak reflection coefficients; and (2) multiple scattering caused by coarse aggregates can mask true interlayer reflections and interfere with image interpretation.

Therefore, accurate interpretation and inversion of GPR images for concrete pavements must take into account the geometry, spatial distribution, and electromagnetic properties of coarse aggregates. Introducing a triphasic medium model allows for a more realistic simulation of radar wave propagation and energy distribution within composite materials. This enables clear differentiation between structural reflections (e.g., interfaces or voids) and material-related scattering (e.g., aggregates). By properly addressing aggregate-induced signal interference, the accuracy and physical consistency of data interpretation can be significantly improved. This provides a solid theoretical and technical basis for precise detection and quantitative assessment of pavement defects.

## 6. Conclusions

This study developed a triphasic concrete pavement model using the FDTD method to simulate electromagnetic responses under various structural and voiding conditions. The main findings are as follows:

(1) The heterogeneous distribution of coarse aggregates induces strong scattering during wave propagation, generating pseudo-hyperbolic echoes in B-scan images and complicating GPR interpretation with an increased risk of false positives.

(2) Aggregate-induced scattering limits the minimum detectable thickness of voids. At 1 GHz, the detection threshold is about 10 mm; thinner voids produce weak reflections that are easily masked by noise.

(3) Void geometry and continuity strongly affect radar imaging. Continuous elongated voids appear as stable planar reflections, whereas discrete small-scale voids generate localized hyperbolic patterns.

In summary, this work integrates high-fidelity triphasic modeling with multi-frequency field validation to clarify the electromagnetic response of pavement voids. The proposed model captures the influence of concrete heterogeneity on radar propagation, providing a robust basis for distinguishing structural reflections from aggregate-induced scattering. These results enhance the accuracy of void detection, reduce interpretation errors, and support the broader application of GPR in non-destructive pavement evaluation and intelligent diagnosis of defects in complex composite systems.

Nevertheless, some limitations should be noted. The model does not fully account for factors such as material moisture variability or large-scale field heterogeneity, which may influence radar responses in practical applications. Future work will aim to refine the model by incorporating these factors and validating its applicability under more diverse conditions.

## Figures and Tables

**Figure 1 sensors-25-05713-f001:**
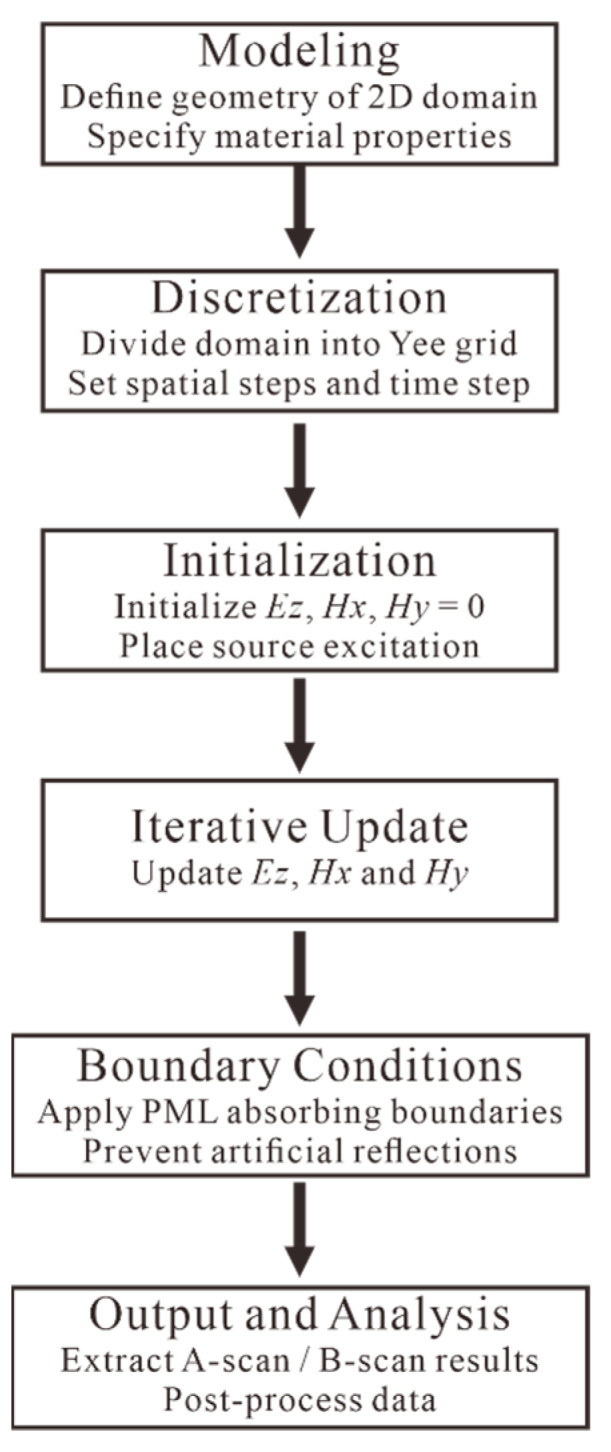
Block diagram of FDTD implementation steps.

**Figure 2 sensors-25-05713-f002:**
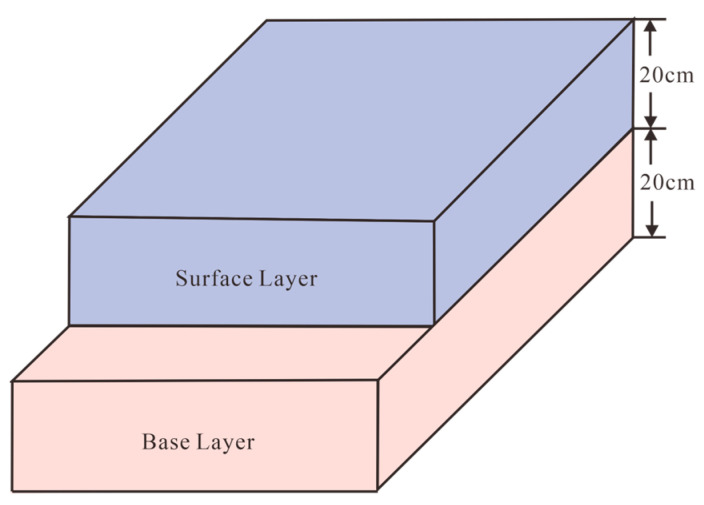
Schematic composition of pavement structure.

**Figure 3 sensors-25-05713-f003:**
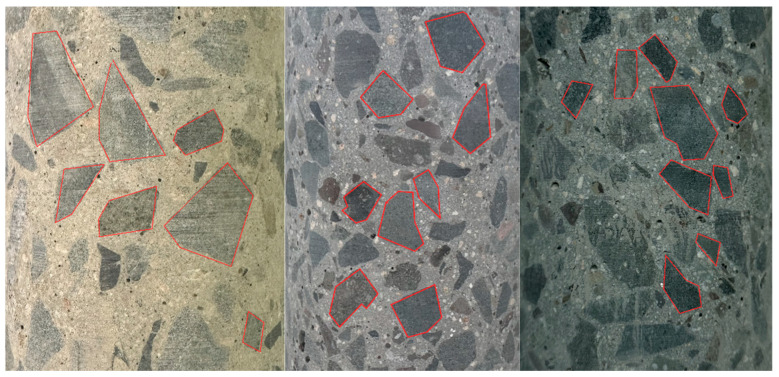
2D image of real concrete aggregates.

**Figure 4 sensors-25-05713-f004:**
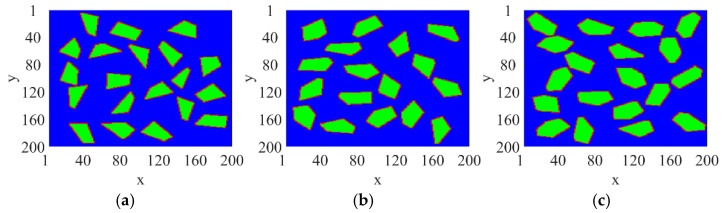
Simulated 2D aggregates. (**a**) Quadrilateral aggregate; (**b**) Pentagonal aggregate; (**c**) Hexagonal aggregate.

**Figure 5 sensors-25-05713-f005:**
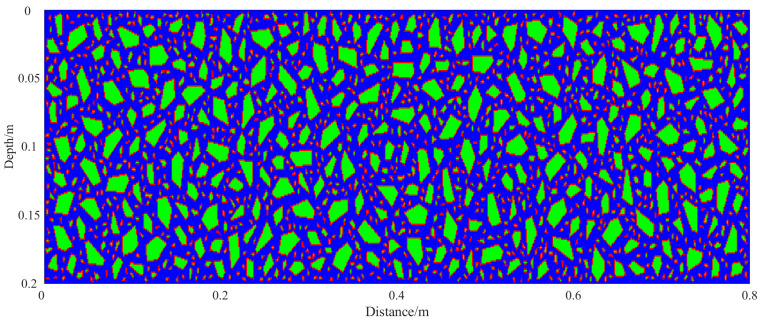
Simulated two-dimensional section of the surface layer block of concrete pavement. Green, red, and blue represent aggregates, ITZ, and mortar, respectively.

**Figure 6 sensors-25-05713-f006:**
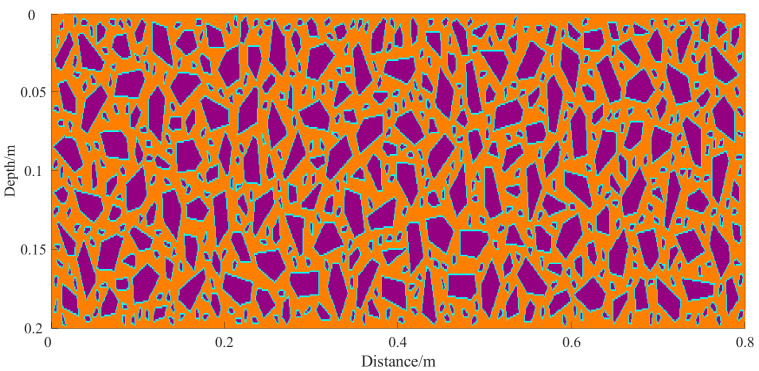
Simulated two-dimensional section of the base layer block of concrete pavement. Purple, cyan, and orange represent aggregates, ITZ, and mortar, respectively.

**Figure 7 sensors-25-05713-f007:**
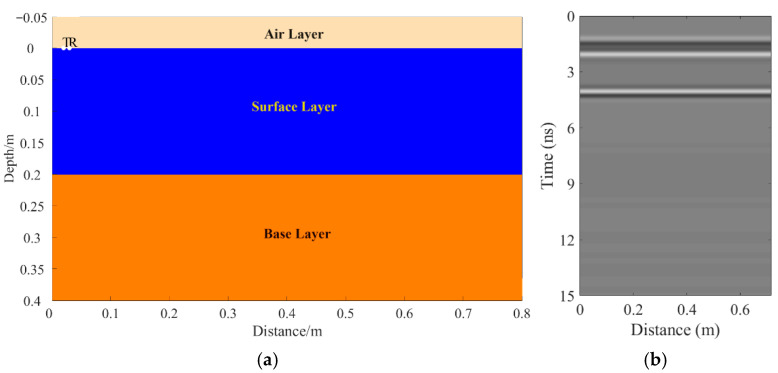
Model one: The surface and base layers are assumed to be homogeneous concrete. (**a**) Homogeneous concrete model; (**b**) B-scan image.

**Figure 8 sensors-25-05713-f008:**
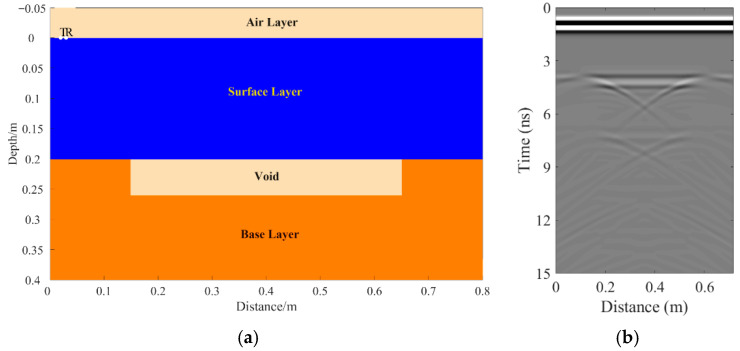
Model two: The surface and base layers are homogeneous concrete with a void in the base layer. (**a**) Homogeneous concrete model with an embedded void; (**b**) B-scan image.

**Figure 9 sensors-25-05713-f009:**
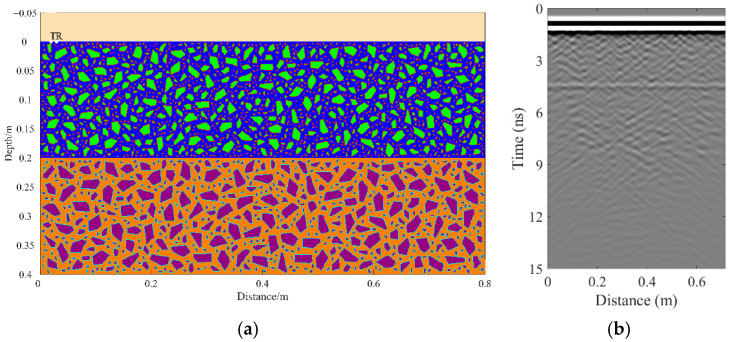
Model three: Concrete composed of aggregates, mortar, and ITZ. (**a**) Three-phase concrete model; (**b**) B-scan image.

**Figure 10 sensors-25-05713-f010:**
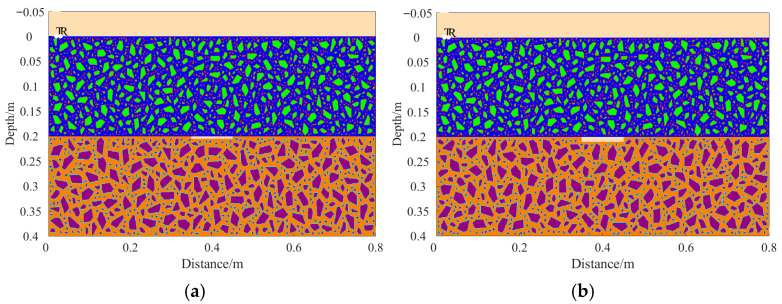
Three-phase concrete models with different void thicknesses. (**a**) Model four: Void size 100 × 5 mm; (**b**) Model five: Void size 100 × 10 mm.

**Figure 11 sensors-25-05713-f011:**
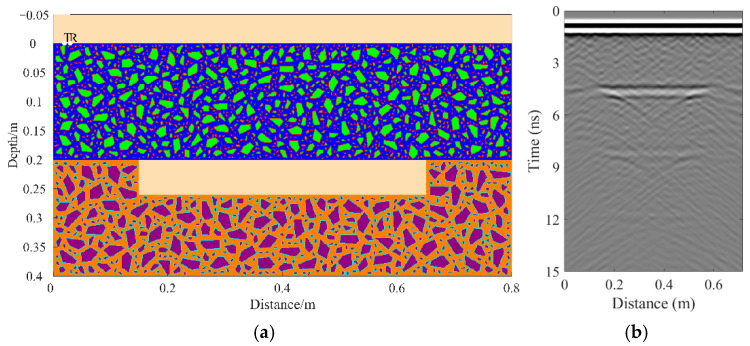
Model six: Three-phase concrete model with continuous voids. (**a**) Continuous void model in three-phase concrete; (**b**) B-scan image.

**Figure 12 sensors-25-05713-f012:**
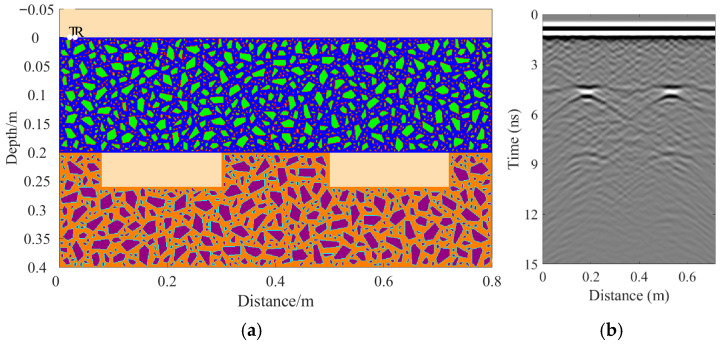
Model seven: Three-phase concrete model with discontinuous voids. (**a**) Discontinuous void model in three-phase concrete; (**b**) B-scan image.

**Figure 13 sensors-25-05713-f013:**
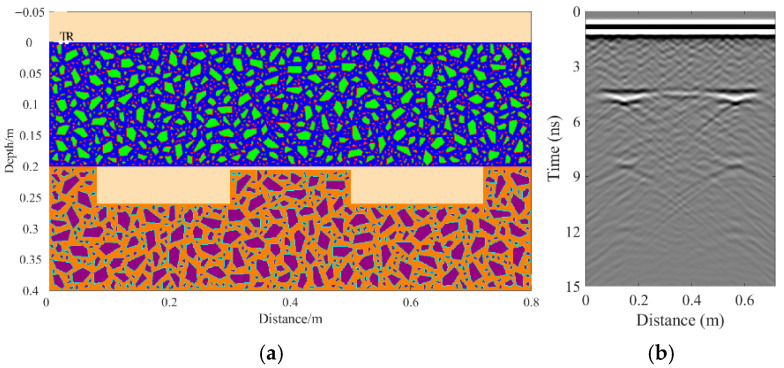
Model eight: Three-phase model with thin voids of varying thickness. (**a**) Three-phase concrete model with 5 mm thin void; (**b**) B-scan image.

**Figure 14 sensors-25-05713-f014:**
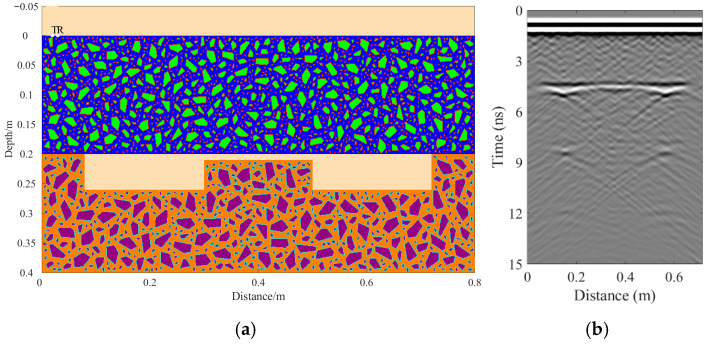
Model nine: Three-phase model with thin voids of varying thickness. (**a**) Three-phase concrete model with 10 mm thin void; (**b**) B-scan image.

**Figure 15 sensors-25-05713-f015:**
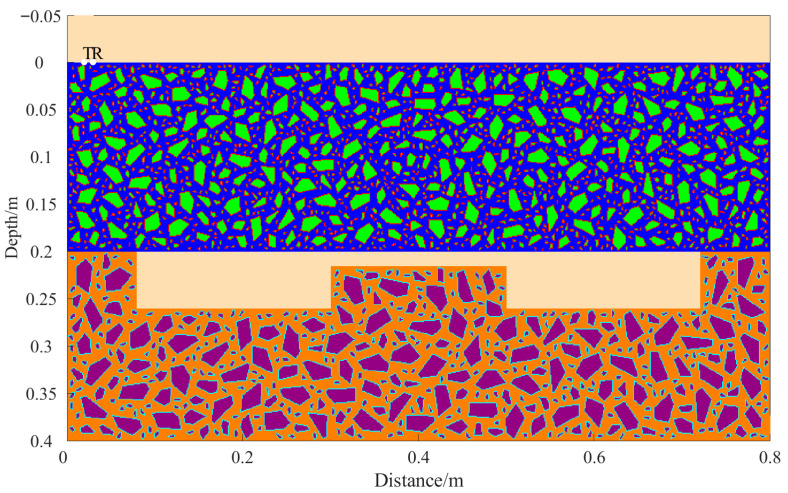
Model ten: Three-phase concrete model with 15 mm thin void.

**Figure 16 sensors-25-05713-f016:**
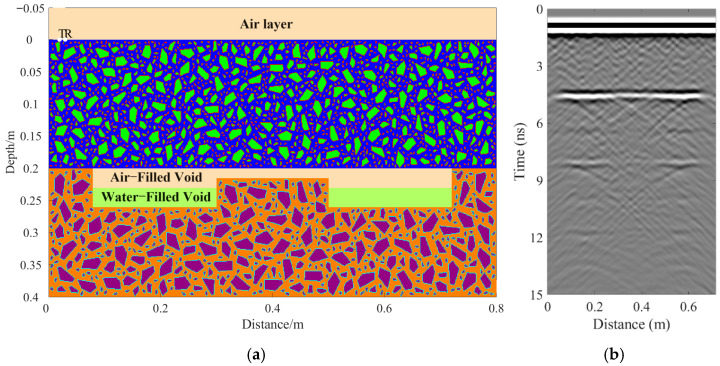
Model eleven: Water-filled three-phase concrete model. (**a**) Three-phase concrete model with a 15 mm void filled with a 30 mm water layer; (**b**) B-scan image.

**Figure 17 sensors-25-05713-f017:**
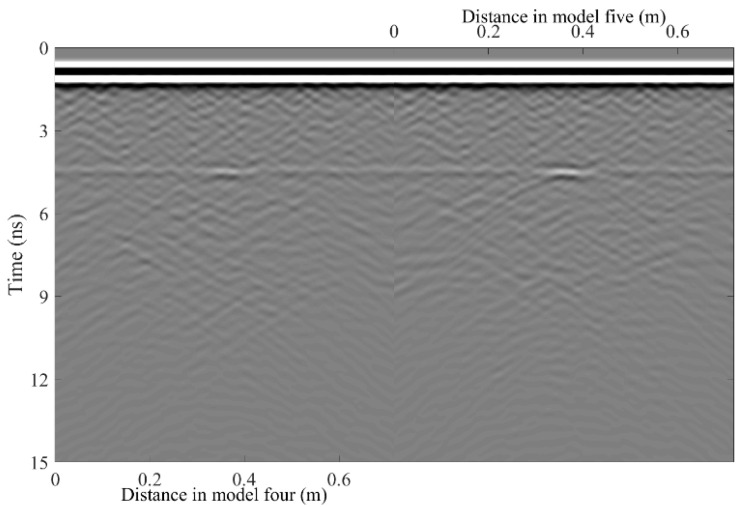
B-scan images of model four and model five.

**Figure 18 sensors-25-05713-f018:**
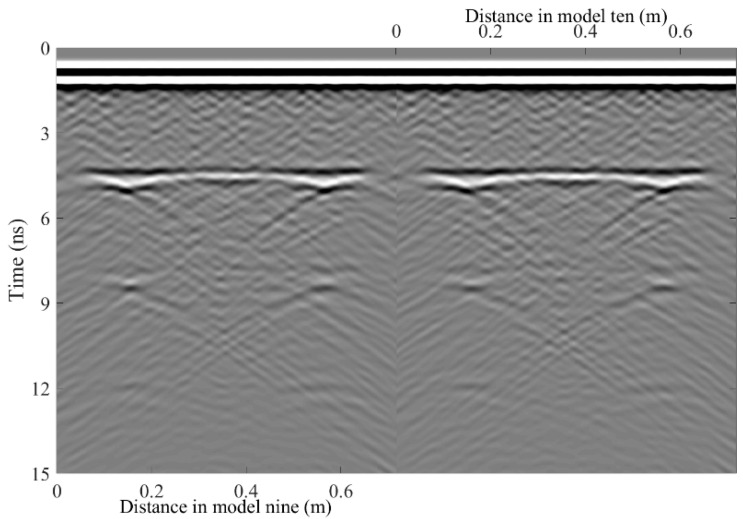
B-scan images of model nine and model ten.

**Figure 19 sensors-25-05713-f019:**
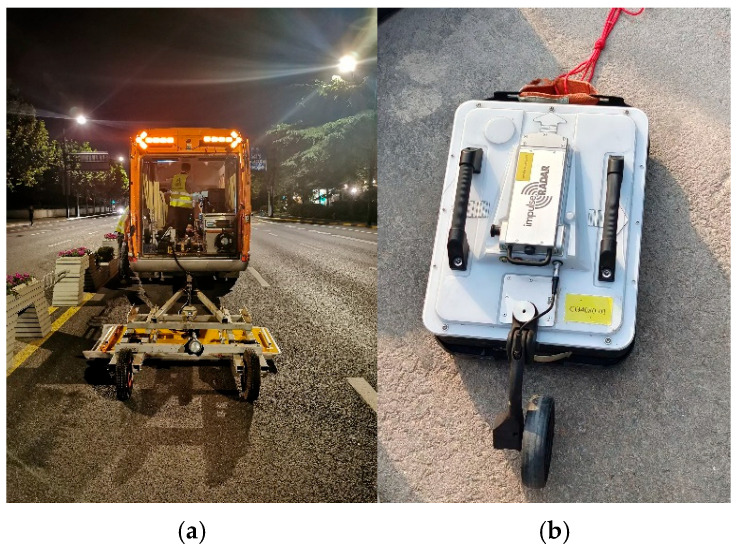
Radar systems used in field testing. (**a**) 3D-Radar GeoScope™ MK IV system; (**b**) ImpulseRadar CO1760 dual-frequency system.

**Figure 20 sensors-25-05713-f020:**
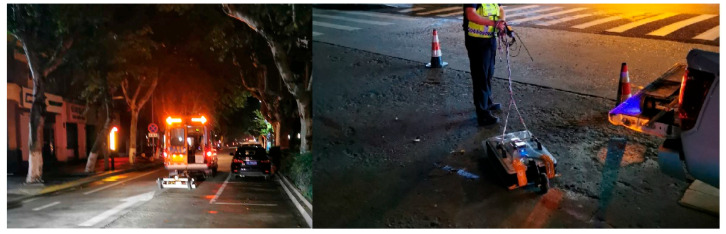
On-site deployment of 3D and 2D radar systems.

**Figure 21 sensors-25-05713-f021:**
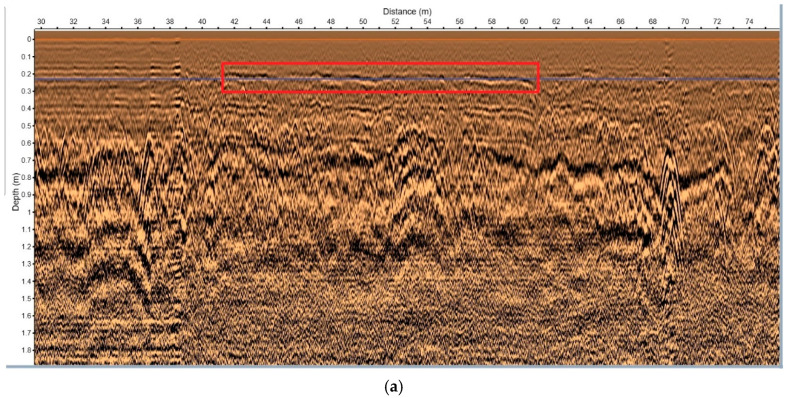
3D GPR survey of an in-service concrete pavement. (**a**) B-scan; (**b**) C-scan.

**Figure 22 sensors-25-05713-f022:**
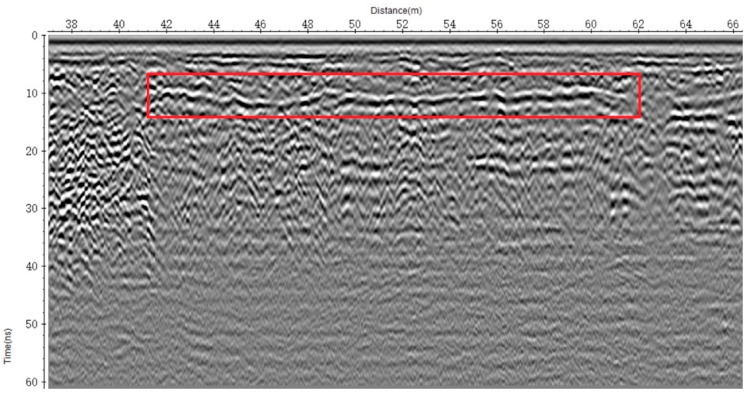
2D GPR survey of the same concrete pavement (B-scan).

**Figure 23 sensors-25-05713-f023:**
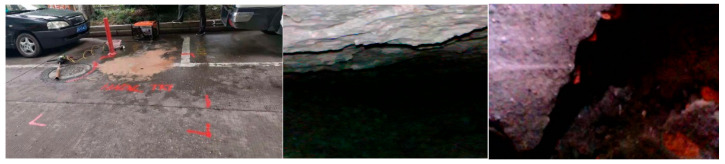
Field verification and internal view of the detected void.

**Table 1 sensors-25-05713-t001:** Mix design of the surface layer.

Aggregate 0–5 mm (kg/m^3^)	Aggregate 5–10 mm (kg/m^3^)	Aggregate 10–20 mm (kg/m^3^)	Aggregate 20–25 mm (kg/m^3^)	Cement (kg/m^3^)	Admixture (kg/m^3^)	Water (kg/m^3^)	*w/c* ratio	Sum (kg/m^3^)
600	141	515	752	403	6	105	0.52	2522

**Table 2 sensors-25-05713-t002:** Mix design of the cement-stabilized base layer.

Material Composition	Aggregate Size	Aggregate 20–30 mm	Aggregate 10–20 mm	Aggregate 5–10 mm	Aggregate 0–5 mm
	Proportion (%)	18	40	15	27
Cement Content (%)	4.5				

**Table 3 sensors-25-05713-t003:** Numerical simulation models structural parameters for cement concrete pavement.

Layer	Thickness (m)	Material	Relative Dielectric Constant
Surface	0.2	Aggregate	5
		Mortar	8
		ITZ	7
Base	0.2	Aggregate	5
		Mortar	11
		ITZ	10

**Table 4 sensors-25-05713-t004:** Technical specifications of the XG1820 antenna.

Antenna Model	Width (mm)	Frequency Range (MHz)	Number of Channels	Channel Spacing (mm)	Effective Scan Width (mm)
DXG1820	1800	800	20	75	1500

## Data Availability

Data will be made available on request.

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
