# Peer review of "Study on the Ground-Penetrating Radar Response Characteristics of Pavement Voids Based on a Three-Phase Concrete Model"

_sensors, 2025, doi:10.3390/s25185713_

Round 1

Reviewer 1 Report

Comments and Suggestions for Authors

Dear Authors, the topic of the paper is interesting, but how you depicted your research was not well done. The simulations were made using a FDTD approach, but it is not clear if you managed them with your codes or using a commercial software (i.e. gprMax). I suppose that the sections of the Pavement structure were coming from your code, but it was not explained and it is not so clear.

Moreover, the comparison between the simulation results and the real case results was not so clear. You highlighted some good comparison between the simulated and real results, but on your data it is not so clear. Therefore, your conclusions do not reflect the results obtained.

In the attached file, there are detailed comments and suggestions.

Comments on the Quality of English Language

The quality of the english is good

Reviewer 2 Report

Comments and Suggestions for Authors

The manuscript presents a high-fidelity three-phase concrete pavement model using the Finite-Difference Time-Domain (FDTD) method to investigate electromagnetic responses under different voiding conditions. The results are well-supported by both simulations and experimental validation. The study addresses an important challenge in ground-penetrating radar (GPR) applications for concrete pavement inspection, particularly the influence of aggregate-induced scattering on void detection. Overall, the manuscript is well-organized, the methodology is sound, and the findings are significant for improving GPR data interpretation. I recommend minor revision before acceptance. The following points should be addressed to further improve clarity and completeness:

  1. The manuscript provides a broad review of the relevant literature; however, the discussion could be strengthened by adding a more detailed summary of the key findings from previous studies. Highlighting the main results and their relevance to the current research would improve the context and clarity of the work.
  2. The description of the aggregate distribution method could benefit from additional detail. To facilitate replication of the experimental conditions, it would be helpful to include an explicit explanation of how the randomness of the aggregate distribution is achieved.
  3. For completeness, please expand on the description of the admixtures listed in Table 1.
  4. The conclusion section could be further improved by briefly outlining the limitations of the proposed model.

Reviewer 3 Report

Comments and Suggestions for Authors

Dear authors,

Overall, the study is interesting and can be published with minor changes.

The main remark is as follows. It is necessary to initially describe what type or types of GPR are being studied in the research. This will help the reader understand the content of the study more easily.

It is known that GPR are divided into three main classes by the type of emitted signal [1]:

  • Impulse, emitting a short pulse similar in shape to one period of a sinusoid
  • Continuous wave with frequency change law of different types
  • Holographic, usually multi-frequency, which are designed to record three-dimensional images of subsurface object in the probing plane [2, 5].

This should be mentioned both in the abstract and in the introduction.

[1] D.J. Daniels, Surface-Penetrating Radar, London, U.K.: IEE, pp. 300, 1996, ISBN 0852968620

[2] He Zhihua, Liu Tao, Song Xiaoji, Chen Cheng, Jin Guanghu, Huang Chunlin and Su Yi, A holographic subsurface imaging radar with high plane resolution: principle and application, Journal of Physics: Conference Series, Volume 2887, 20th International Conference on Ground Penetrating Radar, 23/06/2024 - 27/06/2024, doi: 10.1088/1742-6596/2887/1/012020

[3] Alexei Popov, Igor Prokopovich, Vladimir Kopeikin, and Dmitry Edemskii, “Synthetic aperture approach to microwave holographic image improvement,” in Days on Diffraction, 2015, pp. 192–197

[4] C. L. Windsor, L. Capineri, and T. D. Bechtel. "Buried object classification using holographic radar." Insight-Non-Destructive Testing and Condition Monitoring 54.6, 2012, pp. 331-339.

[5] Hui Wang, Shiyou Wu, Ling Huang, Guangyou Fang, Xin liu, Sparse Array-based Synthetic Spectrum Imaging Technique, International Conference on Ground Penetrating Radar (GPR), June 2018, doi: 10.1109/ICGPR.2018.8441561
